# Fair Kernel Regression through Cross-Covariance Operators

**Adrián Pérez-Suay**                                     *Adrian.Perez@uv.es*
*Image Processing Laboratory (IPL)*
*Universitat de València*

**Paula Gordaliza**                                     *pgordaliza@bcamath.org*
*Basque Center for Applied Mathematics*
*Universidad Pública de Navarra*

**Jean-Michel Loubes**                                 *loubes@math.univ-toulouse.fr*
*Institut de Mathématiques de Toulouse*
*University Toulouse 3*

**Dino Sejdinovic**                                 *dino.sejdinovic@adelaide.edu.au*
*School of Computer and Mathematical Sciences*
*University of Adelaide*

**Gustau Camps-Valls**                                     *Gustau.Camps@uv.es*
*Image Processing Laboratory (IPL)*
*Universitat de València*

**Reviewed on OpenReview:** *https://openreview.net/forum?id=MyQ1e1VQQ3*

## Abstract

Ensuring fairness in machine learning models is a complex problem from both a formulation and implementation perspective. One sensible criterion for achieving fairness is Equalised Odds, which requires that subjects in protected and unprotected groups have equal true and false positive rates. However, practical implementation is challenging. This work proposes two ways to address this issue through the conditional independence operator. First, given the output values, it is used as a fairness measure of independence between model predictions and sensitive variables. Second, it is used as a regularisation term in the problem formulation, which seeks optimal models that balance performance and fairness concerning the sensitive variables. To illustrate the potential of our approach, we consider different scenarios. First, we use the Gaussian model to provide new insights into the problem formulation and numerical results on its convergence. Second, we present the formulation using the conditional cross-covariance operator. We anticipate that a closed-form solution is possible in the general problem formulation, including in the case of a kernel formulation setting. Third, we introduce a normalised criterion of the conditional independence operator. All formulations are posed under the risk minimisation principle, which leads to theoretical results on the performance. Additionally, insights are provided into using these operators under a Gaussian Process setting. Our methods are compared to state-of-the-art methods in terms of performance and fairness metrics on a representative set of real problems. The results obtained with our proposed methodology show promising performance-fairness curves. Furthermore, we discuss the usefulness of linear weights in the fair model to describe the behaviour of the features when enforcing fairness over a particular set of input features.

## 1 Introduction

Machine learning systems are increasingly deployed in many applications with important societal, economic, or environmental implications. Individuals and society are affected, so handling algorithmic bias to obtain

fairness guarantees has become an important research topic. The term *bias* generally refers to an inclination or prejudice for or against one person or group based on their characteristics and need not have negative connotations. In machine learning, bias can exist in many shapes and forms and can be introduced at any stage in the model development pipeline (Fazelpour and Danks, 2021; Pessach and Shmueli, 2022; Besse et al., 2021). When such bias is related to the information in specific characteristics involving sensitive information, the concept of *algorithmic fairness* is raised. In this sense, an algorithm is considered fair if it makes predictions that do not favour or discriminate against certain individuals or groups based on sensitive characteristics. This definition, however, needs to be properly formalised. Therefore algorithm design, implementation, and deployment need a formal and concrete definition of *favouring* or *disadvantaging* a certain population subgroup. Such quantification of favour or discrimination is one of the fundamental points of contention where ethical issues come into play. Artificial Intelligence (AI) regulation is beginning to be formalised, and it is vitally important that the scientific community is part of this. Indeed, the European Union has recently released the "Ethics guidelines for trustworthy AI" report where it is stated that unfairness and biases must be avoided[1].

The definition of fairness is an elusive question and the first cause of controversy in the field. In general, fairness objectives can be categorised into individual and group fairness. On the one hand, *individual fairness* (Dwork et al., 2012; Joseph et al., 2016; Kusner et al., 2017; Kim et al., 2018; Heidari et al., 2018; Joseph et al., 2018) try to achieve "similar treatment to similar individuals". On the other hand, *group or statistical fairness* focuses on reducing inequalities at a group level, where groups may be defined using sensitive variables such as race, gender, age, or disabilities. The objective is expressed through a measure of statistical independence between the variables involved in the learning process. This approach is the most popular in the literature because of its ease of application to any data distribution (Calders and Verwer, 2010; Kamishima et al., 2012; Kleinberg et al., 2016; Hardt et al., 2016; Chouldechova, 2017; Zafar et al., 2017; Donini et al., 2018; Agarwal et al., 2018; Williamson and Menon, 2019). The first proposal along this line is the *demographic parity* (DP) (Kamiran and Calders, 2009; Kamishima et al., 2011; 2012; Pérez-Suay et al., 2017), which requires the outcome to be independent of the sensitive variables. However, it can impair the ultimate utility we want to achieve (Zhao and Gordon, 2019). In light of this, when the ground truth is available, which is plausible in the supervised learning setup, this independence can be relaxed on the true value of the target. This metric is called *Equalised odds* (EO) (Hardt et al., 2016) and requires the true positive and false positive rates to be the same across different groups.

Obtaining statistical fairness, either DP or EO has been mainly addressed by (i) preprocessing the data to remove sensitive dependencies explicitly (Kamiran and Calders, 2009; Luo et al., 2015; Ristanoski et al., 2013; Gordaliza et al., 2019) or (ii) modifying classification rules incorporating the fairness constraints somehow (Pedreshi et al., 2008; Ruggieri et al., 2010). Furthermore, to a lesser extent, (iii) post-processing the outputs from the learning algorithm with additional minimisation constraints to guarantee fairness can also be found in the literature (Wei et al., 2021). Most methods in the first group seek *fair representation learning*, i.e. achieving fairness through finding an optimal way to preprocess the data and map it into a latent space where all information about the sensitive variables is removed. After such preprocessing, standard machine techniques are employed to build predictive models. Examples of these methods include Zemel et al. (2013); Kamiran and Calders (2012); Adebayo and Kagal (2016); Calmon et al. (2017); Madras et al. (2018); Song et al. (2018); Zhang et al. (2018). Down-weighting or removing sensitive features has also been proposed (Zeng et al., 2016), yet it is often insufficient as related variables may still enter the model (Besse et al., 2021).

The second group (ii) consists of empirical risk minimisation (ERM) approaches that incorporate fairness constraints in the learning process using restrictions or an additional penalty term. From a theoretical point of view, if we denote by $\mathcal{L}$ a loss function, a fair algorithm can be achieved by restricting the loss minimisation over a general class $\mathcal{F}$ of algorithms to a subclass $\mathcal{F}_{\text{Fair}}$ satisfying a chosen fairness criteria. Statistical consistency results for the quantification of the loss in the accuracy of an algorithm under fairness constraints have been the topic of many authors, including Donini et al. (2018) or del Barrio et al. (2020). In the latter, authors introduced the term *price for fairness* for the difference between the risk in the fair class and the general Bayes risk (achieved for the best, but possibly unfair, algorithm), namely $\mathcal{E}(\mathcal{F}_{\text{Fair}}) =$

---

[1]Digital Strategy in Europe - Ethics AI.

$\inf_{f \in \mathcal{F}_{\text{Fair}}} \mathbb{E}(\mathcal{L}(f)) - \inf_{f \in \mathcal{F}} \mathbb{E}(\mathcal{L}(f))$. It is essential to realise that awareness of sensitive information conditions the learning scenario and, consequently, the definition of such a fair class. We propose in section 3.2 a bias-unaware kernel regression setting where the sensitive information is known and it is decided not to be an input to the algorithm and to be used only to impose fairness constraints. In this way, we obtain fair predictors both in the sense of DP (section 3.3) and EO (sections 3.4,3.5). Additionally, we propose an extension to the bias-aware framework in the Appendix, where such sensitive characteristics of the individuals are not observed but assumed to be present in the input data and solve it for the particular case of fair Gaussian models in section 4.

The price for DP has been widely studied in the regression setting, e.g. including mini-max results for computing $\mathcal{E}(\mathcal{F}_{\text{DP}}) = \inf_{f \in \mathcal{F}_{\text{DP}}} \mathbb{E}(\mathcal{L}(f)) - \inf_{f \in \mathcal{F}} \mathbb{E}(\mathcal{L}(f))$, where $\mathcal{F}_{\text{DP}}$ detones the fair class of algorithms satisfying the DP criteria (Gouic et al., 2020; Chzhen et al., 2020). However, there are few contributions to controlling such a quantity $\mathcal{E}(\mathcal{F}_{\text{DP}})$ in a general nonlinear setting. This is mainly due to the challenge of obtaining fair models, which could be characterised as the optimal model in the least-squares sense. Fortunately, the fairness regularisation setting (Kamishima et al., 2012) allowed us to provide a closed-form solution for the general nonlinear setting (Pérez-Suay et al., 2017). In this framework, we precisely obtain an upper bound (in Thm. 3.7) on the difference between the fair estimator and the unrestricted RKHS regressor. On the other hand, it has been shown that obtaining algorithmic outcomes accomplishing EO is generally challenging to formulate and, more notably, to optimise (Woodworth et al., 2017). That is, controlling $\mathcal{E}(\mathcal{F}_{\text{EO}})$, where $\mathcal{F}_{\text{EO}}$ denotes the fair class of algorithms satisfying the EO criteria, is not feasible in general. As an exception, it is possible to obtain a closed-form solution to the minimisation problem in the case of linear models. However, this becomes infeasible in a general nonlinear setting. To address this issue, we propose imposing EO through a measure of conditional dependence via conditional cross-covariance operators defined on reproducing Kernel Hilbert spaces (Fukumizu et al., 2004) as an additional regulariser of the risk minimisation problem (2). This allows us also to obtain upper bounds provided in corollary 3.9, and with a normalised version of the conditional cross-covariance operator in corollary 3.12.

**Contributions.** The formulation of the fair learning problem proposed here is distinct and differs from previous proposals. The paper provides four main contributions in a bias-unaware setting: (1) control of the loss between the best DP fair regressor (Pérez-Suay et al., 2017) and the unconstrained RKHS regressor; (2) the use of the conditional independence measure given by the conditional cross-covariance operator (Fukumizu et al., 2008) as an EO measurement in the general kernel regression setting; (3) a closed-form solution to the EO fair optimal predictor in such context; as well as (4) an upper bound for the loss between the best EO fair regressor and the best unconstrained one. It is also worth mentioning our proposal of a probabilistic treatment of this case with Gaussian Processes (GPs) in the Appendix. Moreover, in a bias-aware setting, two additional contributions are (5) a closed-form solution to the EO fair optimal predictor under the linear regression model for multidimensional $X$ and $S$; (6) a proposal to extend previous contributions (1-4) to the bias-aware kernel setting in the Appendix.

**Structure of the paper.** The remainder of the article is structured as follows. First, in section 2, we present the penalised empirical risk minimisation framework for the fair learning supervised problem. Then, the general kernel setting is studied in section 3 in this regularisation framework using two different fairness penalties to the loss minimisation problem: the HSIC operator as the DP penalty in section 3.3; the conditional cross-covariance operator as the EO penalty in section 3.4, as well as a normalised version of the latter in section 3.5. Later in section 4, we solve the fair minimisation problem under the standard regression model. Finally, we give empirical evidence of performance in both synthetic and real-life experiments 5. Appendices contain technical details, a probabilistic treatment based on GPs, and information about the data and implementation issues. A working implementation, demos and code snippets are available at https://www.uv.es/pesuaya/data/code/2023_FACIL.zip.

## 2 Problem setting

The crucial aim of fair learning is designing procedures to remove or control the influence of sensitive information on the forecast that could lead to unfair decisions. Let us denote a general class of ML algorithms by $\mathcal{F}$. We consider a supervised fair learning setting where an algorithm $f \in \mathcal{F}$ is designed to learn the

relationships between input variables and a target variable $Y \in \mathbb{R}$. Here we consider both input variables multidimensional, namely $X \in \mathcal{X} \subset \mathbb{R}^d$, $S \in \mathcal{S} \subset \mathbb{R}^s$, with $d, s \geq 1$.

The learning setting is different depending on the algorithm's awareness of the sensitive information. On the one hand, in most fundamental problems, the input can be separated into a sensitive part $S \in \mathcal{S}$ and a non-sensitive, but a proxy of the former, part $X \in \mathcal{X}$. In this bias-unaware scenario where $\mathcal{F} = \{f : \mathcal{X} \mapsto \mathbb{R}\}$, even if the algorithm $f$ is trained only from $X$, it may be able to learn the information in $S$, resulting in potentially biased outcomes $\hat{Y} = f(X)$ as represented in Figure 1. The other possible formulation of the problem is to consider a bias-aware setting where the algorithm $f(\tilde{X})$ is trained from the complete input data $\tilde{X} = (X, S)$. This framework is helpful in fields such as Econometrics, where it is common to assume the presence of an unobserved sensitive variable $S$ within the input data $\tilde{X}$, as described, for instance, by Fabris et al. (2022) or Bird et al. (2019), and references therein.

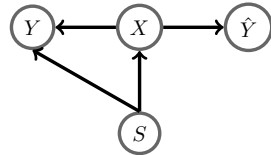

Figure 1: Bias-unaware scenario.

From a mathematical point of view, group fairness consists of statistical independence between the predictions $\hat{Y}$ and the sensitive variable $S$ (del Barrio et al., 2020). The most stringent and common definition is *demographic parity* (DP), which requires the independence between the outcome and the protected attribute $\hat{Y} \perp\!\!\!\perp S$. When the ground truth is available, which is plausible in the supervised learning setup, the *equalised odds* (EO) criteria relaxes this independence on the true value of the target, namely $\hat{Y} \perp\!\!\!\perp S \mid Y$. Hence, building fair algorithms through an ERM-based approach turns the minimisation of the risk $\mathbb{E}(\mathcal{L}(\hat{Y})) = \mathbb{E}[(Y - \hat{Y})^2]$ into a multi-objective problem which requires a trade-off between the model's accuracy and fairness. While in some fields of application, it is desirable to ensure the highest possible level of fairness, in others, including health care or criminal justice, performance should not be decreased since the decisions would have serious implications for individuals and society. Establishing such a trade-off has been one of the important challenges in the ML community in the last years, and a wide array of approaches has been proposed in the literature (Dwork et al., 2012; Chouldechova, 2017; Zafar et al., 2017; Gordaliza et al., 2019; Chiappa et al., 2020). The majority of the works focus on a bias-unaware learning setting where the risk minimisation problem is

$$\inf_{f \in \mathcal{F}} \mathbb{E}(\mathcal{L}(f)) = \inf_{f \in \mathcal{F}} \mathbb{E}[(Y - f(X))^2], \ \mathcal{F} = \{f : \mathcal{X} \mapsto \mathbb{R}\}. \tag{1}$$

Set the learning sample as i.i.d. observations $(x_1, s_1, y_1), \ldots, (x_n, s_n, y_n)$ of $(X, S, Y)$ drawn from an unknown distribution $\mathbb{P}$. Then, for a given model class $\mathcal{F}$, consider $\widehat{f}_n$ the best model that can be estimated by minimising over $\mathcal{F}$,

$$\widehat{f}_n \in \arg\min_{f \in \mathcal{F}} \left\{ \sum_{i=1}^{n} (y_i - f(x_i))^2 + \lambda \Omega(f) \right\}, \tag{2}$$

where $\Omega(f)$ acts as a regularizer of the predictive function and controls the smoothness and complexity of the model. We denote by $f^*$ the oracle rule

$$f^\star \in \arg\min_{f \in \mathcal{F}} \mathbb{E}_{\mathbb{P}}(\mathcal{L}(f)) + \lambda \Omega(f),$$

which is the best (yet unknown) predictor that could be constructed if the true distribution were known.

Now we want to additionally impose group fairness constraints. For this, we include an additional regularisation term $I(f(X), S)$ into the general problem (2) accounting for independence between the predictive function $f$ and the sensitive variables in $S$ as follows

$$\mathcal{L}_{n,\lambda,\mu}(f) := \sum_{i=1}^{n} (y_i - f(x_i))^2 + \lambda \Omega(f) + \mu I(f(x_i), s_i), \tag{3}$$

where $\lambda, \mu \geq 0$. The solution $\hat{f}_{n,\lambda,\mu} \in \arg\min_{f \in \mathcal{F}} \mathcal{L}_{n,\lambda,\mu}(f)$ is the optimal fair predictor. Notice that the above formulation (3) allows us to impose the two notions of group fairness. Indeed, $I(f(x_i), s_i)$ would be a measure of dependence between $f$ and $S$ in the case of imposing DP, or $f|Y$ and $S$ in the case of EO.

In this work, we propose a regularisation approach for (1) to obtain a solution for a bias-unaware fair predictor in a general non-linear setting under both fairness criteria. To do this, we measure the independence via two different covariance operators: the well-known Hilbert-Schmidt Independence Criterion (HSIC) for DP and the conditional cross-covariance operator (Fukumizu et al., 2008) for EO. Additionally, we propose in the Appendix an extension to the bias-aware setting where minimisation (1) would be done from functions $f \in \mathcal{F} = \{f : \mathcal{X} \times \mathcal{S} \mapsto \mathbb{R}\}$. In particular, we address this case for Gaussian models and obtain a closed-form solution for the optimal fair predictor and a lower bound for the minimal excess risk.

## 3 Fair kernel regression through dependence regularisation

Several authors have considered the framework of penalised regression either to achieve fairness or to provide guarantees for independence. In Kamishima et al. (2012), the authors propose to use a logistic loss and a version of the mutual information estimator. In the framework of RKHS regression, Pérez-Suay et al. (2017) quantified the dependence using the Hilbert-Schmidt Independence Criterion (HSIC) based on the norm of the particular cross-covariance operator on the corresponding RKHS. In a seminal paper (Gretton et al., 2005), closed-form solutions to the kernel ridge regression are obtained even when the sensitive variable is multidimensional.

### 3.1 Notations and preliminaries

We shall first introduce some notations on kernel theory that will be used in the following sections.

**Definition 3.1** (Metric spaces, kernels and adjoint operators). *Let $\mathcal{X}$ be a metric space and assume $K : \mathcal{X} \times \mathcal{X} \to \mathbb{R}$ is a measurable kernel with associated RKHS $\mathcal{H}_K$ and canonical feature map $\phi : \mathcal{X} \to \mathcal{H}_K$. Suppose $K$ is bounded, i.e. $\sup_{x \in \mathcal{X}} K(x,x) =: \kappa^2 < \infty$. By $S_\phi : \mathcal{H}_K \hookrightarrow L^2(\mathcal{X}, P_X)$ we denote the inclusion. The adjoint operator $S_\phi^* : L^2(\mathcal{X}, P_X) \to \mathcal{H}_K$ is given by*

$$S_\phi^* g = \int_X g(x)\phi(x) \, dP_X(x),$$

*for any $g \in L^2(\mathcal{X}, P_X)$.*

We further introduce some relevant operators needed in the following:

**Definition 3.2** (Covariance operator). *The* non-centred *operator $T_\phi := S_\phi^* S_\phi : \mathcal{H}_K \to \mathcal{H}_K$ that is given by*

$$T_\phi f = \int_X \langle f, \phi(x) \rangle_K \phi(x) \, dP_X(x),$$

*and the* integral *operator $L_\phi := S_\phi S_\phi^* : L^2(\mathcal{X}, P_X) \to L^2(\mathcal{X}, P_X)$, given by*

$$L_\phi g = \int_X g(x)\langle \phi(\cdot), \phi(x) \rangle_K \, dP_X(x) \, .$$

It can be shown that both operators $T_\phi$ and $L_\phi$ are positive and trace class, hence compact, satisfying

$$\|T_\phi\| \leq tr(T_\phi) \leq \kappa^2 \, , \quad \|L_\phi\| \leq tr(L_\phi) \leq \kappa^2 \, ,$$

where $tr(\cdot)$ denotes the trace of an operator. Note that for any $f \in \mathcal{H}_K$ we have

$$\|\sqrt{T_\phi}f\|_K = \|S_\phi f\|_{L^2(P_X)} \, .$$

For given data $\mathbf{x} = (x_1, ..., x_n)$ drawn i.i.d. from $P_X$ we define the *sampling operator* by

$$\hat{S}_\phi : \mathcal{H}_K \to \mathbb{R}^n \, , \quad f \mapsto \hat{S}_\phi f := (f(x_1), ..., f(x_n)) \, .$$

One easily verifies that the adjoint operator $\hat{S}_\phi^* : \mathbb{R}^n \to \mathcal{H}_K$ is given by

$$\hat{S}_\phi^* \mathbf{y} = \frac{1}{n} \sum_{j=1}^n y_j \phi(x_j) \, ,$$

for any $\mathbf{y} = (y_1, ..., y_n) \in \mathbb{R}^n$. With these definitions, we additionally introduce $\hat{T}_\phi : \mathcal{H}_K \to \mathcal{H}_K$ as

$$\hat{T}_\phi := \hat{S}_\phi^* \hat{S}_\phi = \frac{1}{n} \sum_{j=1}^n \phi(x_j) \otimes \phi(x_j)$$

and $\hat{L}_\phi : \mathbb{R}^n \to \mathbb{R}^n$ as

$$\hat{L}_\phi := \hat{S}_\phi \hat{S}_\phi^* = \frac{1}{n} \mathbb{K} \, ,$$

where $\mathbb{K} = (K(x_i, x_j))_{i,j=1,...,n}$ denotes the *kernel matrix*.

## 3.2 Bias-unaware fair kernel regression setting

**Definition 3.3.** *Nonparametric model, feature maps and reproducing kernels Let $\mathcal{X}$ and $\mathcal{S}$ be standard Borel. Consider the non-parametric regression model*

$$Y = f^*(X) + \varepsilon, \tag{4}$$

*where $\mathbb{E}[\varepsilon|X, S] = 0$ and $(X, S, Y) \in \mathcal{X} \times \mathcal{S} \times [-M, M]$ are random variables, distributed according to some unknown measure $\mathbb{P}$. We consider the corresponding feature maps*

$$\phi : \mathcal{X} \to \mathcal{H}_K, \quad \psi : \mathcal{S} \to \mathcal{H}_{K_S},$$

*into separable RKHSs $(\mathcal{H}_K, \langle \cdot, \cdot \rangle_K)$ and $(\mathcal{H}_{K_S}, \langle \cdot, \cdot \rangle_{K_S})$ with associated reproducing kernels*

$$K(x, x') = \langle \phi(x), \phi(x') \rangle_K, \quad K_S(s, s') = \langle \psi(s), \psi(s') \rangle_{K_S}$$

*see e.g. (Steinwart and Christmann, 2008, Ch. 4).*

In a bias-aware framework, the sensitive information $S$ is given in conjunction with the other characteristics of the individuals $X$ in the data. Therefore, the problem consists in minimising (2) over functions $f : \mathcal{X} \mapsto \mathbb{R}$. It is well-known that the regression function $f^*(x) = \mathbb{E}[Y|X = x] \in L^2(\mathcal{X}, P_X)$ is the unique minimiser of $\mathcal{L}$. Now, given a sample $(\mathbf{x}, \mathbf{s}, \mathbf{y}) := ((x_1, s_1, y_1), ..., (x_n, s_n, y_n))$, we consider the empirical minimisation problem (2) with the usual penalty $\Omega(f) = \|f\|_K^2$. Using the representer theorem (Schölkopf et al., 2001), the minimiser $\hat{f}_{n,\lambda} = \arg\min \mathcal{L}_{n,\lambda}(f)$ can be written as

$$\hat{f}_{n,\lambda}(x) = \sum_{i=1}^n \hat{\alpha} K(x, x_i),$$

where model weights $\hat{\alpha}_n$ are estimated by minimising the new loss in the dual $\arg\min \|y - K\alpha\|^2 + \lambda\alpha^T K\alpha$, which has a closed-form solution $\hat{\alpha}_n = (K + \lambda I)^{-1} Y$.

To impose group fairness constraints, we include an additional regularisation term into the general problem (2) accounting for independence between the predictive function $f$ and the sensitive variables in $S$:

$$\mathcal{L}_{n,\lambda,\mu}(f) := \sum_{i=1}^n (y_i - f(x_i))^2 + \lambda\|f\|_K^2 + \mu I(f(x_i), s_i), \tag{5}$$

where $\lambda, \mu \geq 0$, and the predictive function is the result $\hat{f}_{n,\lambda,\mu} \in \arg\min_f \mathcal{L}_{n,\lambda,\mu}(f)$. For the sake of readability, in the following, we omit subindex $n$ and denote the minimiser of (5) by $\hat{f}_{\lambda,\mu}^{\text{DP}}$ or $\hat{f}_{\lambda,\mu}^{\text{EO}}$ depending on the fairness penalty $I$.

**Remark 3.4.** *This formulation of the problem could be extended to a bias-aware framework where the sensitive information is assumed to exist but not observed; that is, it is hidden in the complete input $\tilde{X} = (X, S)$ from which $f$ is trained. See more details at Appendix A.1*

In the following sections, we obtain a solution for a fair predictor in each case by measuring the independence via two different covariance operators, precisely the well-known Hilbert-Schmidt Independence Criterion (HSIC) for DP and the Conditional Cross-covariance Operator (Fukumizu et al., 2008) for EO.

### 3.3 DP-fair kernel regression through HSIC operator

Measuring statistical independence using reproducing kernels has been investigated in Gretton et al. (2005; 2007). We recall some basic definitions and facts that we need in the sequel. In particular, the Hilbert-Schmidt Independence Criterion (HSIC) measures dependence between random variables $X$ on a domain $X$ and $S$ on a domain $S$ with joint distribution $P_{XS}$. Roughly speaking, HSIC measures the distance between an embedding of $P_{XS}$ and the product of the marginals $P_X \otimes P_S$ into an appropriate RKHS.

To this end, let us introduce a useful operator in this work:

**Definition 3.5.** Cross-covariance operator $\mathcal{C}_{SX} : \mathcal{H}_K \to \mathcal{H}_{K_S}$

$$\mathcal{C}_{SX} := \mathbb{E}_{SX}[(\psi(S) - \mu_S) \otimes (\phi(X) - \mu_X)] = \mathbb{E}_{SX}[\psi(S) \otimes \phi(X)] - \mu_S \otimes \mu_X .$$

*Given a sample* $(\mathbf{x}, \mathbf{s}) := ((x_1, s_1), \ldots, (x_n, s_n))$, *the empirical cross-covariance operator* $\widehat{\mathcal{C}}_{\mathbf{sx}} : \mathcal{H}_K \to \mathcal{H}_{K_S}$ *is defined as*

$$\widehat{\mathcal{C}}_{\mathbf{sx}} = \frac{1}{n} \sum_{j=1}^{n} \phi(s_j) \otimes \psi(x_j) - \hat{\mu}_{\mathbf{s}} \otimes \hat{\mu}_{\mathbf{x}} ,$$

*where*

$$\hat{\mu}_{\mathbf{s}} = \frac{1}{n} \sum_{j=1}^{n} \psi(s_j) , \qquad \hat{\mu}_{\mathbf{x}} = \frac{1}{n} \sum_{j=1}^{n} \phi(x_j) .$$

*Then* $\mathrm{HSIC}(P_{SX}, \mathcal{H}_K, \mathcal{H}_{K_S}) := \|\mathcal{C}_{SX}\|_{HS}^2 .$

We solve the minimisation problem by regularised empirical risk minimisation. In particular, fairness as DP is incorporated by adding the additional penalty term $I(f(x_i), s_i) = \|\widehat{\mathcal{C}}_{\mathbf{sx}} f\|_{K_S}^2$, which can detect statistical independence $f \perp\!\!\!\perp S$. A short calculation, after applying an extension of the representer theorem, shows that the minimiser of the above problem (5) is given by

$$\hat{f}_{\lambda,\mu}^{\mathrm{DP}} = (\hat{T}_\phi + \lambda I + \mu \widehat{\mathcal{C}}_{\mathbf{sx}}^* \widehat{\mathcal{C}}_{\mathbf{sx}})^{-1} \hat{S}_\phi^* \mathbf{y} . \tag{6}$$

**Remark 3.6.** *The assumptions required for the representer theorem (Schölkopf et al., 2001) are met in our setting. Firstly, we employ a standard training sample $\{(x_i, y_i)\}_{i=1}^{N} \subset \mathbb{R}^d \times \mathbb{R}$. Secondly, the function used over the regulariser is the square function $g(\|f\|) = \|f\|^2$, which is a strictly increasing real-valued function. Thirdly, since we utilise norms over kernel matrices, the choice of the cost function can be any arbitrary error function.*

*Moreover, the extension of the representer theorem, which allows for the addition of any penalty $I(f)$, is valid if such penalisation depends only on point values $x_i$. Indeed, following the usual proof in Schölkopf et al. (2001), the solution to the minimisation problem can be projected onto the space generated by $\{K(\cdot, x_1), \ldots, K(\cdot, x_n)\}$. Thus, we have $f = f_K + h$, where $f_K = \sum_i \alpha_i K(\cdot, x_i)$ and the function $h$ is orthonormal to the span of the above space. In particular, since $h(x_i) = \langle h(\cdot), K(\cdot, x_i)\rangle$, utilizing the reproducing property, we find $h(x_i) = 0$.*

*This implies that if the penalty $\Omega(f_K + h) \geq \Omega(f_K)$ is equal to or smaller than that of functions where $h = 0$, and consequently, $h = 0$ for those functions as well. Based on this, it can be deduced that the solution to the minimisation problem can be expressed in terms of functions generated by $K(\cdot, x_i)$.*

We want to emphasise that our use of the regulariser $\left\|\hat{\mathcal{C}}_{sx}f\right\|_{\mathcal{H}}^2$ enables us to obtain the optimal estimator in a closed-form solution, which differs from the HSIC definition. The distinction between these two approaches is discussed in Section 3.3 of the manuscript Li et al. (2022). This enables us to obtain closed-form solutions for the fairness-regularised optimisation (EO) formulations derived in the subsequent sections.

The following Theorem controls the fair penalised estimator and the unrestricted RKHS regressor.

**Theorem 3.7.** *For any $\lambda > 0$ and $\mu > 0$, we have almost surely*

$$\|\hat{f}_{\lambda,\mu}^{\mathrm{DP}} - \hat{f}_{\lambda,0}\|_{L^2(P_X)} \leq M\kappa\|\widehat{\mathcal{C}}_{\mathbf{sx}}^*\widehat{\mathcal{C}}_{\mathbf{sx}}\|\frac{\mu}{4\lambda} \ .$$

*The proof of this Theorem is provided in Appendix A.2.*

The following lemma, identified as Lemma 3.8, demonstrates the relationship between the loss incurred by the fairness regressor and that of the unconstrained regressor.

**Lemma 3.8.** *Let be $f^*$ the best predictor then for any $\lambda \geq 0$ and $\mu \geq 0$, we have almost surely*

$$\left\|f^* - \hat{f}_{\lambda,\mu}^{\mathrm{DP}}\right\|_{L^2(P_X)} \leq \left\|f^* - \hat{f}_{\lambda,0}\right\|_{L^2(P_X)} + \left\|\hat{f}_{\lambda,0} - \hat{f}_{\lambda,\mu}^{\mathrm{DP}}\right\|_{L^2(P_X)}.$$

To prove this lemma, we add and subtract the term $\hat{f}_{\lambda,0}$ and subsequently apply the triangle inequality.

### 3.4 EO-fair kernel regression through conditional cross-covariance operator

The hardness of implementing EO as the fairness constraints into statistical methods usually derives from relaxations of that condition, such as Equalised correlations or equality of opportunities. In this work, we directly impose such a fairness condition by considering a conditional independence measure as a penalty term to the usual risk minimisation problem.

Since it is well-known that the optimal fair predictor in the sense of EO, namely $\arg\min_{f\in\mathcal{F}_{\mathrm{EO}}} \mathcal{L}_{n,\lambda}(f)$, is not feasible to compute (Woodworth et al., 2017), we propose alternatively to quantify the fairness regularizer $I(f,X,S) =$ through the conditional cross-covariance operator

$$\mathcal{C}_{fS|Y} = \mathcal{C}_{fS} - \mathcal{C}_{fY}\mathcal{C}_{YY}^{-1}\mathcal{C}_{YS}. \tag{7}$$

The above operator was introduced in Fukumizu et al. (2004) as a measure of conditional independence between random variables. The measure was further formalised later in Fukumizu et al. (2008). This work introduces the conditional independence operator as a fairness regulariser. The EO can be expressed in terms of an extended version of (7) as

$$\hat{Y} \perp\!\!\!\perp S|\ Y \quad \Leftrightarrow \quad \|\mathcal{C}_{fS|Y}\|_{HS}^2 = 0.$$

Therefore, given a sample $(\mathbf{x},\mathbf{s},\mathbf{y}) := ((x_1,s_1,y_1),\ldots,(x_n,s_n,y_n))$, EO is impose by adding $I(f,X,S) = \|\widehat{\mathcal{C}}_{\mathbf{fs}|\mathbf{y}}\|_{HS}^2$ *empirical conditional cross-covariance operator* $\widehat{\mathcal{C}}_{\mathbf{fs}|\mathbf{y}} : \mathcal{H}_K \to \mathcal{H}_{K_S}$ is defined as

$$\widehat{\mathcal{C}}_{\mathbf{fs}|\mathbf{y}} = \widehat{\mathcal{C}}_{\mathbf{fs}} - \widehat{\mathcal{C}}_{\mathbf{fy}}\widehat{\mathcal{C}}_{\mathbf{yy}}^{-1}\widehat{\mathcal{C}}_{\mathbf{ys}}.$$

A short calculation shows that the minimiser of the fairness regularisation problem is given by

$$\hat{f}_{\lambda,\mu}^{\mathrm{EO}} = (\hat{T}_\phi + \lambda I + \mu\widehat{\mathcal{C}}_{\mathbf{fs}|\mathbf{y}}^*\widehat{\mathcal{C}}_{\mathbf{fs}|\mathbf{y}})^{-1}\hat{S}_\phi^*\mathbf{y}.$$

The justification for the above formula is similar to the previous section, and the Remark 3.6 remains valid in this context. Consequently, we can state the following result that controls the loss of the best fair constrained predictor and the best (possibly unfair) unconstrained predictor.

**Theorem 3.9.** *For any $\lambda > 0$ and $\mu > 0$, we have almost surely*

$$\|\hat{f}_{\lambda,\mu}^{\mathrm{EO}} - \hat{f}_{\lambda,0}\|_{L^2(P_X)} \leq M\kappa\|\widehat{C}_{\mathbf{fs}|\mathbf{y}}^*\widehat{\mathcal{C}}_{\mathbf{fs}|\mathbf{y}}\|\frac{\mu}{4\lambda}.$$

*The proof of this theorem follows the proof of the Theorem 3.7.*

Furthermore, we introduce a partially-normalised EO regressor in 3.5.

The following lemma, identified as Lemma 3.10, demonstrates the relationship between the loss incurred by the fairness regressor and that of the unconstrained regressor.

**Lemma 3.10.** *Let be $f^*$ the best predictor then for any $\lambda \geq 0$ and $\mu \geq 0$, we have almost surely*

$$\left\| f^* - \hat{f}_{\lambda,\mu}^{\text{EO}} \right\|_{L^2(P_X)} \leq \left\| f^* - \hat{f}_{\lambda,0} \right\|_{L^2(P_X)} + \left\| \hat{f}_{\lambda,0} - \hat{f}_{\lambda,\mu}^{\text{EO}} \right\|_{L^2(P_X)}.$$

To prove this lemma, we add and subtract the term $\hat{f}_{\lambda,0}$ and subsequently apply the triangle inequality.

We will use the same notation for the matrices associated with the random variables to improve readability. Specifically, let $X \in \mathbb{R}^{n \times d}$ be the matrix of the complete input data (samples $\times$ dimensionality), $S \in \mathbb{R}^{n \times s}$ the sensitive variable ($S$ is treated as a subset of $X$), and $Y$ contains the observed true labels, which can be considered in the multi-output setting being a matrix of outputs. Recall that $S$ is not limited to the one-dimensional case; rather than that, it can consider an arbitrary, finite number of sensitive attributes ($s \geq 1$).

We expand our formulation to the use of kernels in the nonlinear framework. Then, we map the input data to an RKHS $X \to \Phi$, recall that $\Phi\Phi^\top = K$; also map the sensitive attributes as $S \to \Psi$, where $\Psi\Psi^\top = K_S$; and the representer theorem (Schölkopf et al., 2001) states that $\omega = \Phi^\top \alpha$. The solution of the so-called FACIL (Fair Conditional Independence Learning) kernel method can be written as

$$\alpha = \left( K + \lambda I + \mu(I - \Pi_y)^\top K_S(I - \Pi_y)K \right)^{-1} Y, \tag{8}$$

where $\Pi_y = Y(Y^\top Y)^{-1} Y^\top$ is the orthogonal projection; hyperparameters $\lambda$ and $\mu$ need to be tuned by cross-validation or fixed *a priori* to achieve a sensible fairness level.

**Remark 3.11.** *The solution of the FACIL in its linear formulation can be written as:*

$$\omega = (X^\top X + \lambda I + \mu X^\top (I - \Pi_y) SS^\top (I - \Pi_y) X)^{-1} X^\top Y.$$

### 3.5 EO with normalised conditional independence measures

In this section, we formalised a framework for including the cross-covariance operators in the cost function and provided a cross-covariance interpretation of fair learning. We follow this motivation to obtain models through the normalised conditional independence operator

$$\mathcal{V}_{SX|Y} = \mathcal{C}_{SS}^{-1/2}(\mathcal{C}_{SX} - \mathcal{C}_{SY}\mathcal{C}_{YY}^{-1}\mathcal{C}_{YX})\mathcal{C}_{XX}^{-1/2}. \tag{9}$$

The above operator (9) codifies the conditional independence relation of $S \perp\!\!\!\perp X|Y$. However, this normalised operator does not allow us to obtain a closed-form solution in the kernel case (we want to note that it can be reached in the linear model formulation). However, a closed-form solution can be achieved in the kernel case by considering the partially normalised operator as follows:

$$\mathcal{V}_{SX|Y} = \mathcal{C}_{SS}^{-1/2}(\mathcal{C}_{SX} - \mathcal{C}_{SY}\mathcal{C}_{YY}^{-1}\mathcal{C}_{YX}), \tag{10}$$

and introducing it in the fairness regularisation term as $I(f, S, X) = \mathcal{V}_{SX|Y} f$. By considering the corresponding empirical estimator $\widehat{\mathcal{V}}_{\mathbf{s}|\mathbf{y}}$, it can be shown that the minimiser of the fairness regularisation problem is given by

$$\widehat{f}_{\lambda,\mu}^{\text{EO}} = (\hat{T}_\phi + \lambda I + \mu \widehat{\mathcal{V}}_{\mathbf{fs}|\mathbf{y}}^* \widehat{\mathcal{V}}_{\mathbf{fs}|\mathbf{y}})^{-1} \hat{S}_\phi^* \mathbf{y}.$$

**Corollary 3.12.** *For any $\lambda > 0$ and $\mu > 0$, we have almost surely*

$$\|\hat{f}_{\lambda,\mu}^{\text{EO}} - \hat{f}_{\lambda,0}\|_{L^2(P_X)} \leq M\kappa \|\widehat{\mathcal{V}}_{\mathbf{fs}|\mathbf{y}}^* \widehat{\mathcal{V}}_{\mathbf{fs}|\mathbf{y}}\| \frac{\mu}{4\lambda}.$$

*The proof of this corollary follows the proof of the Theorem 3.7.*

The following lemma, identified as Lemma 3.13, demonstrates the relationship between the loss incurred by the fairness regressor and that of the unconstrained regressor.

**Lemma 3.13.** *Let be $f^*$ the best predictor then for any $\lambda \geq 0$ and $\mu \geq 0$, we have almost surely*

$$\left\| f^* - \hat{f}_{\lambda,\mu}^{\mathrm{EO}} \right\|_{L^2(P_X)} \leq \left\| f^* - \hat{f}_{\lambda,0} \right\|_{L^2(P_X)} + \left\| \hat{f}_{\lambda,0} - \hat{f}_{\lambda,\mu}^{\mathrm{EO}} \right\|_{L^2(P_X)}.$$

To prove this lemma, we add and subtract the term $\hat{f}_{\lambda,0}$ and subsequently apply the triangle inequality.

By following this, and through the use of kernel feature maps and empirical estimators of the operator, we obtain the dual weights solution of the so-called NFACIL (Fair normalised Conditional Independence Learning):

$$\alpha = (K + \lambda I + \mu(I - \Pi_y)^\top K_S(n\epsilon I + K_S)^{-1}(I - \Pi_y)K)^{-1}Y.$$

Since the existence of $K_S^{-1}$ is not ensured, we included the regularisation $(\Psi^\top \Psi + n\epsilon I)^{-1}$, $\epsilon$ is chosen as in Smola and Schölkopf (2001), and applied the Woodbury-Morrison matrix inversion lemma $(A + BCD)^{-1}BC = A^{-1}B(C^{-1} + DA^{-1}B)^{-1}$.

**Remark 3.14.** *The solutions of both* (8) *and* (11) *require data centring in Hilbert spaces, which can be done implicitly by kernel matrix centring through $H = I - \frac{1}{n}\mathbb{1}\mathbb{1}^\top$.*

**Remark 3.15.** *The solution of the NFACIL, in its linear formulation, can be written as:*

$$\omega = (X^\top X + \lambda I + \mu \mathcal{V}_{SX|y}^\top \mathcal{V}_{SX|y})^{-1}X^\top y.$$

## 4 Bias-aware Gaussian models with EO fairness constraints

We first focus on the particular setting of Gaussian regression models under EO constraints. In this case, estimating an optimal model which satisfies the fairness constraint can be expressed in closed form. To facilitate the understanding of the subsequent experiments in section 5, and without loss of generality, we state this section with $S$ one-dimensional.

Building fair Gaussian models with $S$ known have already been studied in the literature. Precisely, the majority of the works (Tan et al., 2020; Li et al., 2022) considered the linear normal model $Y = f_{\beta_S}^*(X) + \varepsilon$, where the errors $\varepsilon_1, \ldots, \varepsilon_n$ $i.i.d. \sim \mathcal{N}(0, 1)$ are such that $\mathbb{E}(\varepsilon \mid X, S) = 0$, and the predictor is $f_{\beta_S}^*(X) = \beta_S^T X$, where the parameter $\beta_S \in \mathbb{R}^{p \times 1}$ is different for each group. In other words, $S$ different linear models are built.

However, in some contexts including econometrics (Wooldridge, 2002; Stock, 2002), it is natural to assume that certain sensitive information exists but is not observed. To address this case, we rather consider $S$ unknown and set the following model

$$Y = f_{\beta_0,\beta}^*(X, S) + \varepsilon, \tag{11}$$

where the predictor

$$f_{\beta_0,\beta}^*(X, S) = \beta_0 S + \beta^T X, \ \beta_0 \in \mathbb{R}, \ \beta \in \mathbb{R}^{p \times 1} \tag{12}$$

is a linear combination of the sensitive and non-sensitive attributes (and the errors are as above). Then, the joint distribution of $(X, S, Y)$ is $(p + 2)-$dimensional normal, and we denote the vectors of means and the covariance matrices as follows

$$\mathcal{N}\left(\begin{bmatrix} \mu_X \\ \mu_S \\ \mu_Y \end{bmatrix}, \begin{bmatrix} \Sigma_X & \Sigma_{XS} & \Sigma_{XY} \\ \Sigma_{XS}^T & \Sigma_S & \Sigma_{SY} \\ \Sigma_{XY}^T & \Sigma_{SY}^T & \Sigma_Y \end{bmatrix}\right).$$

We note that EO requires the linear fair predictor to be independent of $S$ conditionally given $Y$, that is

$$f_{\beta_0,\beta}^*(X, S) \perp\!\!\!\perp S \mid Y,$$

which under the normal model assumption is equivalent to the second-order moment constraint

$$Cov(f^*(X, S), S \mid Y) = 0. \tag{13}$$

Hence, seeking for fair linear predictor amounts to obtaining conditions on the coefficients $\beta_0, \beta$ for (13) to hold. If we denote by $A_{SXY} \in \mathbb{R}^{p \times 1}$ the *vector of correction for fairness*

$$A_{SXY} := \frac{\Sigma_{XS}\Sigma_Y - \Sigma_{SY}\Sigma_{XY}}{\Sigma_S\Sigma_Y - \Sigma_{SY}^2}, \tag{14}$$

then the optimal fair *equality of odds* predictor under can be exactly computed as follows, whose proof is in the Appendix A.

**Proposition 4.1.** *Under the normal model* (11), *the optimal EO-fair linear predictor of the form* (12) *is given as the solution to the following optimisation problem to obtain* $\beta_{0,fair}, \hat{\beta}_{fair}$

$$\arg\min_{(\beta_0,\beta)\in\mathcal{F}_{\mathrm{EO}}} \mathbb{E}\Big[(Y - f_{\beta_0,\beta}(X,S))^2\Big]$$

*among the class of parameters* $\mathcal{F}_{\mathrm{EO}} = \{(\beta_0,\beta) \in \mathbb{R} \times \mathbb{R}^p\}$ *such that* $\beta^T(\Sigma_{XS}\Sigma_Y - \Sigma_{SY}\Sigma_{XY}) + \beta_0(\Sigma_S\Sigma_Y - \Sigma_{SY}^2) = 0$. *If moreover* $Y$ *and* $S$ *are non-linearly dependent, it can be exactly computed as*

$$\hat{\beta}_{0,fair} = \hat{\beta}_{fair}^T A_{SXY}$$
$$\hat{\beta}_{fair} = \Sigma_Z^{-1}\Sigma_{ZY},$$

*where*

$$\Sigma_Z = \Sigma_X + \Sigma_S A_{SXY} A_{SXY}^T + A_{SXY}\Sigma_{XS}^T + \Sigma_{XS}A_{SXY}^T,$$
$$\Sigma_{ZY} = \Sigma_{XY} + \Sigma_{SY}A_{SXY}.$$

We consider the case where bias is only due to biased observations while the task itself is fair, and therefore $Y$ and $S$ are not linearly dependent.

**Remark 4.2.** *Note that the previous result can be extended to the case where the sensitive variable* $S$ *is multidimensional by considering in* (14) *the matrix of correction for fairness*

$$A_{SXY} = (\Sigma_{XS}\Sigma_Y - \Sigma_{SY}\Sigma_{XY})(\Sigma_S\Sigma_Y - \Sigma_{SY}^2)^{-1}.$$

It is of interest to quantify the price for fairness $\mathcal{E}(\mathcal{F}_{\mathrm{EO}})$ when imposing $(\beta_0,\beta) \in \mathcal{F}_{EO}$. This will be done in section 5 comparing with the general loss associated with the minimiser

$$(\hat{\beta}_0, \hat{\beta}) := \arg\min_{(\beta_0,\beta)\in\mathbb{R}\times\mathbb{R}^p} \mathbb{E}\Big[(Y - f_{\beta_0,\beta}(X,S))^2\Big]. \tag{15}$$

We observe that condition (13) is generally a weaker constraint in a broader setup, where achieving EO conveys computational challenges as discussed in Woodworth et al. (2017). Authors showed that even in the restricted case of learning linear predictors, assuming a convex loss function, and demanding that only the sign of the predictor needs to be non-discriminatory, the problem of matching false positive (FPR) and false negative (FNR) rates requires exponential time to solve in the worst case. Motivated by this hardness result they also proposed a relaxation of the criterion of EO by a more tractable notion of non-discrimination based on second-order moments. In particular, they introduced the notion of *Equalised correlations*, which indeed is generally a weaker condition than (13), but when considering the squared loss and when $(X, S, Y)$ are jointly Gaussian, it is equivalent (and, subsequently, equivalent to EO). In this case, they provided a characterisation of the fair linear predictor as a solution to a minimisation problem solved in the case of the binary target $Y$, where they also computed the price for such a fair predictor.

Since linear prediction can be seen as the most suitable framework for Gaussian processes, the relaxation of (13) could be justified as the appropriate notion of fairness when we restrict ourselves to linear predictors. Furthermore, linear predictors, especially under kernel transformations, are used in various applications. They thus form a practically relevant family of predictors where one would like to achieve non-discrimination.

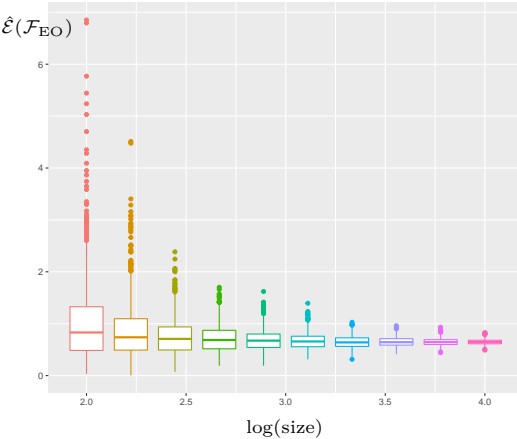

Figure 2: Boxplots of the computations $\hat{\mathcal{E}}(\mathcal{F}_{\mathrm{EO}})$ of the price for fairness under a normal linear model.

In Proposition 4.1, we restrict ourselves to the standard framework in which the computation of the optimal fair predictor is still feasible. Yet, for many distributions and hypothesis classes, there may not exist a non-constant, deterministic, fair predictor. It is worth noting that generalising non-linear Gaussian processes can be a challenging task. Thus, we stress that one can include non-linear basis functions to ensure greater expressivity. Some examples of such basis functions include Random Fourier Features (Rahimi and Recht, 2007) or other known basis functions (Yu et al., 2016; Le et al., 2013). Hence, in the next section, we propose an alternative approach for ensuring EO fairness in a broader setup.

## 5 Experiments

In this section, we provide empirical evidence of the performance of the proposed methods in a set of experiments. Firstly, numerical evidence of convergence of the loss bound in the EO linear regression setting is provided over a simulation set. Secondly, we study the trade-off between error rates and fairness in the proposed cross-covariance metric; the results cover six databases. Thirdly, an empirical comparison of the weights behaviour in the linear model evaluation.

### 5.1 Loss bound EO linear regression

We show results in simulations aimed at estimating the minimal excess risk when imposing EO in the normal linear regression framework, that is, the price for fairness

$$\mathcal{E}(\mathcal{F}_{\mathrm{EO}}) = \left| \mathcal{L}\left( f_{(\hat{\beta}_{0_{fair}}, \hat{\beta}_{fair})} \right) - \mathcal{L}\left( f_{(\hat{\beta}_0, \hat{\beta})} \right) \right|$$

We here consider $S \sim \mathcal{N}(0, 10)$ and $X \in \mathbb{R}^2$, such that

$$X \sim \mathcal{N}\left( \begin{bmatrix} 0 \\ 0 \end{bmatrix}, \begin{bmatrix} 2 & 0 \\ 0 & 3 \end{bmatrix} \right),$$

and $Cov(X_1, S) = Cov(X_2, S) = 0.1$. To simulate the true values of the target $Y$, we have chosen in (12) true parameters $\beta_0^* = 0.1$ and $\beta^* = (2, 1)$.

The results of $1,000$ replications of the experiment, therefore $1,000$ computations of the empirical price for fairness $\hat{\mathcal{E}}(\mathcal{F}_{\mathrm{EO}})$, are shown in Fig. 5.1 taking particularly a log-linearly spaced sequence between 2 and 4 of length 10. In all cases, we varied parameters to the maximum range possible such that the models did not go into numerical errors. In addition, the average minimal excess risk $\overline{\mathcal{E}}(\mathcal{F}_{\mathrm{EO}})$ and its standard deviation as the sample size increases are included in Table 1. We observe that the estimation converges as the averaged value stabilises and the standard deviation numerically decreases.

Table 1: Average minimal excess risk $\overline{\mathcal{E}}(\mathcal{F}_{\mathrm{EO}})$ and standard deviation $\mathrm{sd}(\hat{\mathcal{E}}(\mathcal{F}_{\mathrm{EO}}))$ as sample size increases in a logarithmically spaced grid ranging from $10^2$ to $10^4$ samples.

| Size | 100 | 167 | 278 | 464 | 774 | 1292 | 2154 | 3594 | 5995 | 10000 |
|------|-----|-----|-----|-----|-----|------|------|------|------|-------|
| $\overline{\mathcal{E}}(\mathcal{F}_{\mathrm{EO}})$ | 1.02 | 0.75 | 0.71 | 0.69 | 0.67 | 0.67 | 0.65 | 0.65 | 0.65 | 0.65 |
| $\mathrm{sd}(\hat{\mathcal{E}}(\mathcal{F}_{\mathrm{EO}}))$ | 0.81 | 0.54 | 0.35 | 0.26 | 0.20 | 0.16 | 0.12 | 0.09 | 0.07 | 0.05 |

## 5.2 EO in open datasets

The second set of experiments uses four real datasets (over six considered protected variables) to demonstrate the empirical behaviour of the developed methods. The dataset selection has been made based on the relevance in the field and the broader studies using them. In particular, we consider: 1) the Adult income dataset (Dua and Graff, 2017), 2) the Communities and Crime (Redmond, 2009) (C&C), 3) the National Longitudinal Survey of Youth (Bureau of Labor Statistics, 2019) (NLSY), and 4) the Compas recidivism risk score data (Larson et al., 2016).

Experimental results reveal the trade-off between error rate and fairness through the root-mean-squared error (RMSE) and in EO terms through the conditional independence measure. All the reported results are an average of 25 independent trials. We split data into training, validation and test independent sets. We fix the size of the training set to $N = 600$ samples, the size of the validation set to 100 samples, and the test set to 2000 samples, or the remainder available. Recall that our methods can handle multiple, possibly continuous, protected attributes. We have included Gaussian Process formulations of FACIL and NFACIL methods to provide a complete comparison. GP versions are used to work closer to risk minimisation methods; curves are not too long due to numerical problems reported with the $\mu$ parameter. For this reason, Figure 3[top row] shows the results of the Adult dataset by using $s = 1$ considering the ethnic origin and gender as the sensitive variables independently, as well as deemed together ($s = 2$) in the last column. In the comparison, we have also considered other state-of-the-art methods that handle multiple protected attributes, particularly those proposed in Pérez-Suay et al. (2017); Tan et al. (2020). In the second row, the results of the remaining datasets (2-4) illustrate the good behaviour of our proposal. Specially paradigmatic is the C&C case, where the normalised FACIL-GP consistently reduces conditional independence without increasing. Our approach generally achieves greater accuracy for a given level of fairness.

## 5.3 Linear model sensitivity in weights terms

In this section, we explore the behaviour of the fair DP against EO models unveiled in this work. In particular, we focus on the linear models of the linear-Fair Learning (Pérez-Suay et al., 2017) and the linear-FACIL method.

Fairness conditions have been set to enforce independence along ethnic origin ($9-$th feature) in the Adult database. Figure 4 illustrates the averaged absolute differences between linear-Fair Learning weights' model ($w_{FL}$) and our linear-FACIL weights' model ($w_{FA}$) proposal against least squares (LS) weights' models ($wLS$). As can be seen, the change most notably remains in the enforced feature in both methods. Moreover, the seems to have a more extensive (not significant) difference against the LS weights model.

In light of this, Table 2 summarises a t-test of the hypothesis that both distributions come with equal means. The null hypothesis ("men are equal") cannot be rejected at the 5% significance level for 9 of 14 features. The remaining five appear in the table and indicate that the null hypothesis can be rejected at the 5%

Table 2: Features from Adult database where a t-test of $|w_{LS} - w_{FL}|$ and $|w_{LS} - w_{FA}|$ rejects equal means at the 5% significance level.

| # feature | 1 | 2 | 3 | 11 | 13 |
|-----------|---|---|---|----|----|
| $p$-value ($< 0.05$) | 1.70e-04 | 3.78e-05 | 5.79e-04 | 8.70e-04 | 1.48e-05 |

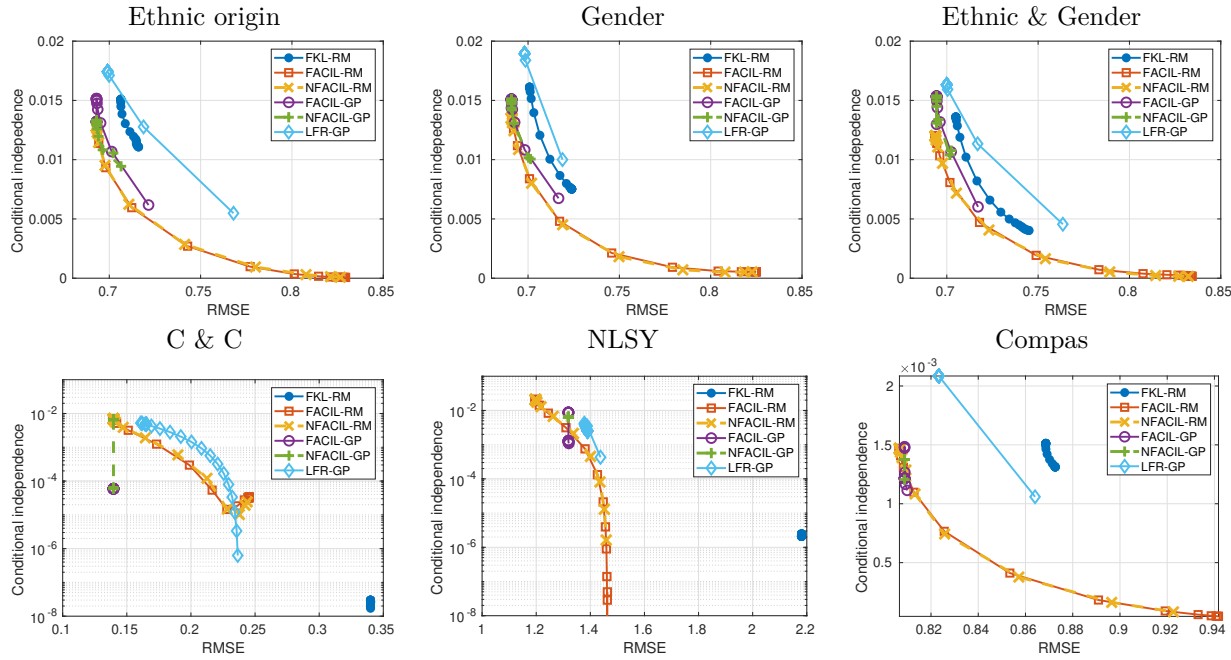

Figure 3: Results on Adult income dataset considering single and multiple sensible variables (first row). Communities and Crime (C & C), National Longitudinal Survey of Youth (NLSY), and the Compas dataset are in the second row.

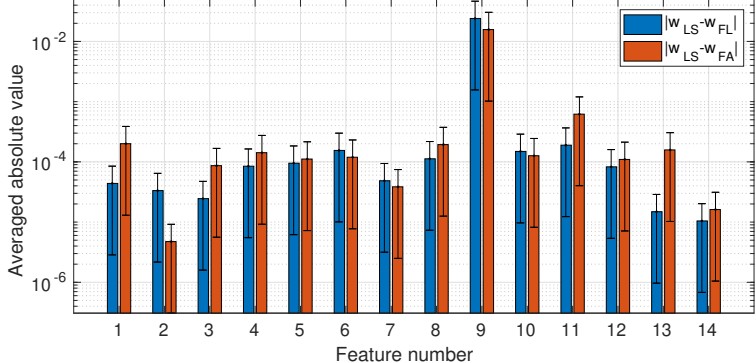

Figure 4: Averaged absolute value of weights subtraction between linear models in the Adult dataset when enforcing fairness with $9-$th feature (ethnic origin). Involved models weights are: Least Squares ($w_{LS}$), Fair Learning ($w_{FL}$), and linear-FACIL ($w_{FA}$).

## 6  Conclusions

We introduced and theoretically motivated a method for learning fair regression models in different settings. Firstly, we focused on the more straightforward case, but also valuable for many practical applications, of a linear Gaussian model. An optimal fair linear predictor satisfying EO is obtained as a function of the model's parameters in this setting. Moreover, we provided simulations that verify the price convergence for fairness. Secondly, an empirical risk minimisation family of methods and the most notable solutions in the linear and kernel cases. Furthermore, the Appendix A provides additional extensions to the Gaussian Process regression details. Along with the proposed methods, two operators have been used to give an EO model. Firstly, the conditional cross-covariance operator is used as an EO fairness measure and a regularisation parameter—secondly, a partially-normalised version of the conditional cross-covariance operator. The use of

those operators reaches closed-form solutions in all cases, providing a straightforward way to develop novel fair methodologies.

We apply our approach to obtain fair models, demonstrating competitive empirical performance on several datasets relative to state-of-the-art methods. Regarding the experimental setup, the fairness parameter $\mu$ allows systemically vary along the fairness-accuracy trade-off. An optimal hyperparameter is problem-specific and should be selected a priori. Importantly, this strictly needs comparable models, which is ensured by introducing the partial normalisation of the operator.

Future work involves developing more scalable algorithms using randomised approximations and providing fairness conditions to other models like neural networks. This could, unfortunately, trade accuracy for the essential properties of the linear kernel and the probabilistic GP treatment. In the future, we also plan to study the effect of the regulariser on the confidence intervals for the predictions and the feature ranking (eventually through ARD kernels) derived from our FACIL-GP models. Finally, there are intrinsic links between fairness, causal inference and physical consistency, and these connections can offer new insights about algorithmic solutions in these fields.

### Acknowledgments

Adrián Pérez-Suay and Gustau Camps-Valls thank Fundación BBVA for their support through the project "Causal inference in the human-biosphere coupled system (SCALE)".

Paula Gordaliza work was supported by the Elkartek program under the Grants 3KIA project (KK-2021/00123) and also by the Basque Government through the BERC 2022-2025 program and by the Ministry of Science and Innovation: BCAM Severo Ochoa accreditation CEX2021-001142-S / MICIN / AEI / 10.13039/501100011033.

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

## A    Appendix

**Remark A.1.** *The problem could be considered in a more general framework where $S$ does not belong to $X$. Assuming $\mathbb{E}_X[\sqrt{K(X,X)}] < \infty$ and $\mathbb{E}_S[\sqrt{K_S(S,S)}] < \infty$, the tensor kernel $K \otimes K_S$ expressed as $(K \otimes K_S)((x,s),(x',s')) = K(x,x')K_S(s,s')$ defines an RKHS $\mathcal{H}_{K \otimes K_S}$ which can be compactly embedded into $L^2(\mathcal{X} \times \mathcal{S}, P_{XS})$.*

*More precisely, we let*

$$T := T_{\psi \otimes \phi} : \mathcal{H}_{K \otimes K_S} \hookrightarrow L^2(\mathcal{X} \times \mathcal{S}, P_{XS}) \tag{16}$$
$$f \to Tf$$

*with*

$$(Tf)(x,s) = \langle f, (K \otimes K_S)((x,s), \cdot) \rangle_{K \otimes K_S} .$$

*In this case, the problem is to minimize over functions $g : \mathcal{X} \times \mathcal{S} \mapsto \mathbb{R}$*

$$\mathcal{L}(g) = \mathbb{E}[(Y - g(X,S))^2] .$$

*It is well known that the regression function $g^*(x,s) = \mathbb{E}[Y|(X,S) = (x,s)] \in L^2(\mathcal{X} \times \mathcal{S}, P_{XS})$ is the unique minimiser of $\mathcal{L}$. In addition, we will assume our model to be* well-specified, *that is, $g^*$ can be written as*

$$g^*(x,s) = \langle f^*, (K \otimes K_S)((x,s), \cdot) \rangle_{K \otimes K_S} ,$$

*for some $f^* \in \mathcal{H}_{K \otimes K_S}$.*

*Given a sample $(\mathbf{x}, \mathbf{s}, \mathbf{y}) := ((x_1, s_1, y_1), ..., (x_n, s_n, y_n))$ we consider the empirical minimization problem*

$$\mathcal{L}_{n,\lambda}(f) := \frac{1}{n} \sum_{j=1}^{n} (f(x_j, s_j) - y_j)^2 + \lambda \|f\|_{K \times K_S}^2$$

*where $\lambda > 0$ is the regularization parameter. Using the property of the RKHS, the solution to this minimization problem has a solution which can be written as*

$$\hat{f}_\lambda(x,s) = \sum_{j=1}^{n} \omega_j K((x,s), (x_j, s_j))$$

*with.*

$$K((x,s), (x_j, s_j)) = K(x, x_j) \otimes K_S(s, s_j),$$

*where the $\omega_j$'s constitute the solution of the minimization*

$$\min_{\omega} \{ \|y - K\omega\|^2 + \lambda K^T \omega K \}$$

*This leads up us to the closed-form solution*

$$\omega = (K + \lambda I)^{-1} (X \otimes \mathbf{S})^\top y. \tag{17}$$

### A.1    Proofs of section 4

We start recalling some facts about Gaussian random variables.

**Proposition A.2.** *If $(U, V, W)$ are jointly Gaussian, then*

- *Conditional expectation $\mathbb{E}(U|V)$ is linear in $V$ and is given by*

$$\mathbb{E}(U|V) = \mathbb{E}(U) + \Sigma_{U,V} \Sigma_V^{-1} (V - \mathbb{E}(V))$$

- *Conditional covariance $\Sigma_{(U,V)|W}$ does not depend on $W$ and is given by*

$$\Sigma_{(U,V)|W} = \Sigma_{U,V} - \Sigma_{U,W}\Sigma_W^{-1}\Sigma_{U,W}^T$$

**Proof of Proposition 4.1.** In the particular normal model, this independence means that the elements in positions $(1,2)$ and $(2,1)$ of the covariance matrix of random vector $(f(X_u, S), S \mid Y)$ are exactly zero. Therefore, the class of fair predictors is written as

$$\mathcal{F}_{\text{EO}} := \{f : \mathcal{X} \times \mathcal{S} \to R^d : Cov(f(X_u, S), S \mid Y) = 0\} \tag{18}$$

More precisely, the previous condition can be written in terms of the covariances of $(X_u, S, Y)$ and the coefficients $(\beta_0, \beta)$ of the linear model (11). Observe that the joint distribution of the random vector $(f_{\beta_0,\beta}(X_u, S), S, Y)$ is

$$\begin{bmatrix} \beta_0 S + \beta^T X_u \\ S \\ Y \end{bmatrix} \sim \mathcal{N}\left( \begin{bmatrix} \beta_0 \mu_S + \beta^T \mu_{X_u} \\ \mu_S \\ \mu_Y \end{bmatrix}, \begin{bmatrix} \Sigma_1 & \Sigma_{12} \\ \Sigma_{12}^T & \Sigma_Y \end{bmatrix} \right),$$

where

$$\Sigma_1 = \begin{bmatrix} \beta_0^2 \Sigma_S + \beta^T \Sigma_{X_u}\beta + 2\beta_0\beta^T\Sigma_{X_u S} & \beta_0\Sigma_S + \beta^T\Sigma_{X_u S} \\ \beta_0\Sigma_{SY} + \beta^T\Sigma_{X_u Y} & \Sigma_S \end{bmatrix} \in \mathbb{R}^{2\times2},$$

$$\Sigma_{12} = \begin{bmatrix} \beta_0\Sigma_{SY} + \beta^T\Sigma_{X_u Y} \\ \Sigma_{SY} \end{bmatrix} \in \mathbb{R}^{2\times1}.$$

Hence, from Proposition A.2, we know that

$$Cov(f(X_u, S), S \mid Y) = \Sigma_1 - \frac{1}{\Sigma_Y}\Sigma_{12}\Sigma_{12}^T.$$

Substituting the expressions above for $\Sigma_1$ and $\Sigma_2$, we obtain that $g_{\beta_0,\beta} \in \mathcal{F}_{\text{EO}}$ if and only if

$$(\beta_0\Sigma_S + \beta^T\Sigma_{X_u S})\Sigma_Y = \Sigma_{SY}(\beta_0\Sigma_{SY} + \beta^T\Sigma_{X_u Y}).$$

Then the optimal EO-fair predictor in this setting is the solution to the following optimization problem:

$$\left(\hat{\beta}_{0,fair}, \hat{\beta}_{fair}\right) := \arg\min_{(\beta_0,\beta)\in\mathcal{F}_{\text{EO}}} \mathbb{E}\left[(Y - f_{\beta_0,\beta}(X_u, S))^2\right] \tag{19}$$

$$\mathcal{F}_{\text{EO}} = \{(\beta_0, \beta) \in \mathbb{R} \times \mathbb{R}^p \mid \beta^T(\Sigma_{X_u S}\Sigma_Y - \Sigma_{SY}\Sigma_{X_u Y}) + \beta_0(\Sigma_S\Sigma_Y - \Sigma_{SY}^2) = 0\}.$$

We note that the Cauchy-Schwarz inequality and the assumption that $Y$ and $S$ are nonlinearly dependent ensure $\Sigma_S\Sigma_Y - \Sigma_{SY}^2 > 0$. Then we obtain that the class of EO-fair predictors $(\beta_0, \beta) \in \mathcal{F}_{\text{EO}}$ are such that $\beta_0 = \beta^T C_{S,X_u,Y}$, where

$$C_{S,X_u,Y} := \left(\frac{\Sigma_{X_u S}\Sigma_Y - \Sigma_{SY}\Sigma_{X_u Y}}{\Sigma_S\Sigma_Y - \Sigma_{SY}^2}\right) \in \mathbb{R}^{p\times1}.$$

Hence, the optimal EO-fair predictor (19) can be obtained equivalently

$$\hat{\beta}_{fair} = \arg\min_{\beta\in\mathbb{R}^p} \mathbb{E}\left[\left(Y - \beta^\top(X_u + SC_{S,X_u,Y})\right)^2\right].$$

Now if we denote $Z := X_u + SC_{S,X_u,Y}$, it is easy to check that the optimal EO-fair predictor can be exactly computed as

$$\hat{\beta}_{fair} = \Sigma_Z^{-1}\Sigma_{Z,Y}, \text{ where}$$

$$\Sigma_Z = \Sigma_{X_u} + \Sigma_S C_{S,X_u,Y} C_{S,X_u,Y}^T + C_{S,X_u,Y}\Sigma_{X_u S}^T + \Sigma_{X_u S}C_{S,X_u,Y}^T$$

$$\Sigma_{ZY} = \Sigma_{X_u Y} + \Sigma_{SY}C_{S,X_u,Y}.$$

$\square$

## A.2   Proofs of section 3

**Proof of Theorem 3.7.** For any two bounded invertible operators $A, B$ acting on some Hilbert space we have the relation

$$A^{-1} - B^{-1} = A^{-1}(B - A)B^{-1} .$$

This can be proved by expanding the right term as follows $A^{-1}(B - A)B^{-1} = A^{-1}(BB^{-1}) - (AA^{-1})B^{-1} = A^{-1}(I) - (I)B^{-1}$. From this, we easily deduce:

$$(\hat{T}_\phi + \lambda I + \mu \widehat{\mathcal{C}}^*_{\mathbf{sx}} \widehat{\mathcal{C}}_{\mathbf{sx}})^{-1} - (\hat{T}_\phi + \lambda I)^{-1}$$
$$= -\mu (\hat{T}_\phi + \lambda I + \mu \widehat{\mathcal{C}}^*_{\mathbf{sx}} \widehat{\mathcal{C}}_{\mathbf{sx}})^{-1} \widehat{\mathcal{C}}^*_{\mathbf{sx}} \widehat{\mathcal{C}}_{\mathbf{sx}} (\hat{T}_\phi + \lambda I)^{-1} .$$

Moreover, the spectral theorem yields (Theorem 2.7-8 in Kreyszig (1978), page 96), through the polar decomposition $\hat{S}_\phi = U|\hat{S}_\phi|$ (Theorem 2, in Vito et al. (2005)) the following

$$\|(\hat{T}_\phi + \lambda I)^{-1}\| \leq \frac{1}{2\sqrt{\lambda}},$$

and, we define the cross-covariance operator as $\widehat{\mathcal{C}}^*_{\mathbf{sx}} \widehat{\mathcal{C}}_{\mathbf{sx}} = K_S K = S^*_\psi S_\psi S^*_\phi S_\phi$, and $S_\psi$ is the sampling operator through $\psi$. As both $S_\phi, S_\psi$ are bounded operators, we recall the polar decomposition of $S_\phi = U|S_\phi|, S_\psi = V|S_\psi|$. Then,

$$\|(\hat{T}_\phi + \lambda I + \mu \widehat{\mathcal{C}}^*_{\mathbf{sx}} \widehat{\mathcal{C}}_{\mathbf{sx}})^{-1} S^*_\phi\| = \|(S^*_\phi S_\phi + \lambda I + \mu S^*_\psi S_\psi S^*_\phi S_\phi)^{-1} S^*_\phi\| = \|(|S_\phi|^2 + \lambda I + |S_\psi|^2 |S_\phi|^2)^{-1}|S^*_\phi|\| =$$

$$\left\| \sup_{\substack{t \in [0, \|S_\phi\|] \\ l \in [0, \|S_\psi\|]}} \frac{t}{t^2 + \lambda + \mu l^2 t^2} \right\| = \frac{1}{2\sqrt{\lambda}}$$

$$\|(\hat{T}_\phi + \lambda I + \mu \widehat{\mathcal{C}}^*_{\mathbf{sx}} \widehat{\mathcal{C}}_{\mathbf{sx}})^{-1}\| \leq \frac{1}{2\sqrt{\lambda}} .$$

Hence, since $\|S^*_\phi \mathbf{y}\|_K \leq M\kappa$ we get using (21)

$$\|\hat{f}_{\lambda,\mu} - \hat{f}_{\lambda,0}\|_{L^2(\mathcal{X}, P_X)} \leq \|(\hat{T}_\phi + \lambda I + \mu \widehat{\mathcal{C}}^*_{\mathbf{sx}} \widehat{\mathcal{C}}_{\mathbf{sx}})^{-1}(-\mu \widehat{\mathcal{C}}^*_{\mathbf{sx}} \widehat{\mathcal{C}}_{\mathbf{sx}})(\hat{T}_\phi + \lambda I)^{-1})\| \|S^*_\phi \mathbf{y}\|_K \leq M\kappa \|\widehat{\mathcal{C}}^*_{\mathbf{sx}} \widehat{\mathcal{C}}_{\mathbf{sx}}\| \frac{\mu}{4\lambda} .$$

$\square$

## A.3   Summary of the proposed methods

Table 3: Summary of the proposed methods based on the frequentist ERM principle. Method name, operator acting as a regularizer and weights' solution in its corresponding linear ($\omega$) or kernel ($\alpha$) counterpart.

| Method | Operator | Solution |
|---|---|---|
| FACIL | $\Sigma_{S\hat{y}|y}$ | $\omega = (X^\top X + \lambda I + \mu X^\top (I - \Pi_y) S S^\top (I - \Pi_y) X)^{-1} X^\top y$ |
| K-FACIL | $\Sigma_{S\hat{y}|y}$ | $\alpha = (K + \lambda I + \mu (I - \Pi_y) K_S (I - \Pi_y) K)^{-1} y$ |
| NFACIL | $\mathcal{V}_{SX|y} = \Sigma_{SS}^{-1/2} (\Sigma_{SX|y}) \Sigma_{yy}^{-1/2}$ | $\omega = (X^\top X + \lambda I + \mu \hat{\Sigma}_{XX}^{-1/2} \hat{\Sigma}_{SX|y}^\top \hat{\Sigma}_{SS}^{-1} \hat{\Sigma}_{SX|y} \hat{\Sigma}_{XX}^{-1/2})^{-1} X^\top y$ |
| K-NFACIL | $\mathbf{P}_{SX|y} = \Sigma_{SS}^{-1/2} (\Sigma_{SX|y})$ | $\alpha = (K + \lambda I + \mu (I - \Pi_y) K_S (n\epsilon I + K_S)^{-1} (I - \Pi_y) K)^{-1}$ |

## A.4   EO with Gaussian Processes

We can derive a probabilistic version of FACIL using Gaussian processes (Rasmussen and Williams, 2006). Standard regression approximates observations $\{y_i\}_{i=1}^n$ as the sum of some unknown latent function $f(x)$ of the some covariates $\{x_i \in \mathbb{R}^d\}_{i=1}^n$ plus *constant power (homoscedastic)* Gaussian noise, i.e. $y_i = f(x_i) + e_i$

where $e_i \sim \mathcal{N}(0, \sigma_e^2)$. GP regression proceeds in a Bayesian, non-parametric way, to fit observations. For convenience, let us assume the transformed data $\tilde{X} = S^\top (I - \Pi_y)X$ collectively contains all transformed points $\tilde{x} = S^\top (I - \Pi_y)x$. Likewise, we can define the relation explicitly in Hilbert spaces: $\tilde{\Phi} = S^\top (I - 1\Pi_y)\Phi = C\Phi$ so we can compute the corresponding kernel matrix (without explicitly mapping the data or applying the transformation) as $\tilde{K} = \tilde{\Phi}\tilde{\Phi}^\top = CKC^\top$.

Essentially a GP prior is placed on the latent function $f(\tilde{x})$ and a Gaussian prior is used for each latent noise term $e_i$, $f(\tilde{x}) \sim \mathcal{GP}(0, \tilde{K}_\delta(\tilde{x}, \tilde{x}'))$, where $\tilde{K}_\delta(\tilde{x}, \tilde{x}')$ is a covariance function parameterized by $\delta$, and $\sigma_e^2$ is a hyper-parameter that specifies the noise variance so that the vector containing all the hyper-parameters is $\theta = [\delta, \sigma_e]$.

Essentially, a GP is a stochastic process whose marginals are distributed as a multivariate Gaussian. In particular, given the prior $\mathcal{GP}$, samples drawn from $f(\tilde{x})$ at the set of locations $\{\tilde{x}_i\}_{i=1}^n$ follow a joint multivariate Gaussian with zero mean and covariance matrix $\tilde{K}$ with $[\tilde{K}]_{ij} = K_\delta(\tilde{x}_i, \tilde{x}_j)$. Specifically, considering $y = [y_1, \ldots, y_N]^\top$ and $\tilde{X} = [\tilde{x}_1, \ldots, \tilde{x}_N]^\top$, with a GP formulation we can obtain analytically $p(f(\tilde{x})|\tilde{x}, \tilde{X}, y) = \mathcal{N}(f(\tilde{x})|\hat{f}(\tilde{x}), \hat{\sigma}(\tilde{x}))$, where the posterior mean and variance are:

$$\hat{f}(\tilde{x}) = \tilde{K}(\tilde{x}, \tilde{X})^\top (\tilde{K} + \sigma_e^2 I_N)^{-1} y,$$
$$\hat{\sigma}(x) = \tilde{K}(\tilde{x}, \tilde{x}) - \tilde{K}(\tilde{x}, \tilde{X})(\tilde{K} + \sigma_e^2 I_N)^{-1}\tilde{K}(\tilde{x}, \tilde{X})^\top.$$

The FACIL-GP uses the kernel $\tilde{K} = CKC^\top$, which can be readily computed and optimized with marginal log-likelihood maximization. GP regression and the kernel ridge regression solution for the predictive mean are identical (Kanagawa et al., 2018). However, with a GP formulation, we can directly control the matrix $\tilde{K}$ by the choice of the kernel function $\tilde{K}_\delta(\tilde{x}_i, \tilde{x}_j)$, infer the model parameters by maximum log-likelihood, and obtain the posterior predictive variance $\hat{\sigma}(x)$, not just point-wise estimates for the prediction.

Given a sample $(\mathbf{x}, \mathbf{s}, \mathbf{y}) := ((x_1, s_1, y_1), ..., (x_n, s_n, y_n))$ we consider minimising

$$\min_{f \in \mathcal{H}_K} \hat{\mathcal{L}}(f) \tag{20}$$

with

$$\hat{\mathcal{L}}(f) := \frac{1}{n}\sum_{j=1}^n (f(x_j) - y_j)^2 + \lambda\|f\|_K^2 + \mu\|\hat{\mathcal{C}}_{\mathbf{sx}}f\|_G^2 \, ,$$

where $\lambda > 0$, $\mu > 0$ are the regularisation parameters. A short calculation shows that the minimiser of the above problem is given by

$$\hat{f}_{\lambda,\mu} = (\hat{T}_\phi + \lambda I + \mu\hat{\mathcal{C}}_{\mathbf{sx}}^* \hat{\mathcal{C}}_{\mathbf{sx}})^{-1}\hat{S}_\phi^* \mathbf{y} \, . \tag{21}$$

