# OpenReview forum: "Fair Kernel Regression through Cross-Covariance Operators"
_TMLR — Accepted by TMLR_

### Review · Reviewer_yBfZ · 2023-04-22

**Summary Of Contributions:**

The paper considers the problem of imposing fairness constraints to the empirical risk minimization problem.

They indirectly do so by penalizing the model via proxies that measure how independent appropriate values are: e.g. in the case of demographic parity, independence between the predicted value hat{y} and the group membership s.

In the kernel regression setting, they are able to derive a closed form solution that is close to the the true optimal solution for the above fairness regularized problem for demographic parity and equalized odds (Section 3).

In section 4, they study the setting where the model can take the demographic group information as an input; they call this Bias-aware models.

Finally, they have some experiments that complement their theoretical results above.

**Audience:**

Yes

**Broader Impact Concerns:**

I don’t have any broader impact concern for this paper.

**Claims And Evidence:**

Yes

**Requested Changes:**

I will describe the changes that would be helpful in light of the weakness described above
(1) Presentation: I think the writing can be improved by initially trying to describe the high level goal/motivation of the paper. Also, it would be helpful to describe the connection between the results at a high level so that the readers can more easily organize the results in their mind as they are reading.

(2) I think definition environments should be used to formally state important definitions. Also, trying to avoid overloading notations and making sure all the notations are always defined before using them would greatly improve the paper.

(3) In section 3 & 4, it would be helpful to have a little longer discussion about why the suggested fairness regularization term are good proxies for the original fairness goal, even when they are not exactly 0. For instance, for equalized odds, can it be said that the the smaller this regularization term, the smaller the difference in the true/false positive rates always? Similarly, instead of plotting the conditional independence in the Pareto curves, it would be more helpful to trace out the difference in true/false positive rates for equalized odds and the difference in the positive prediction rates for demographic parity. If the answer to the question is true, then it seems like the shape of these plots should stay the same.

**Strengths And Weaknesses:**

Strengths:
I think the main strength of the paper is that it derives the actual closed form solution of the problem they consider in various settings. A lot of the in-processing methods come up with an optimization method for solving the unfairness regularized empirical risk minimization problems, but having a closed form solution gives rise to easy computation of these fair models.



Weakness:
(1) The presentation of the paper can be improved significantly.

First, it was very hard to understand the results of the paper in a structured manner. For instance, in the abstract, it tries to describe things in terms of the different operators (conditional independence, normalized conditional independence operators) without explaining these concepts, and I don’t think the readers would be familiar with them.

I think it would be helpful to initially describing these results in a much more succinct manner without necessarily trying to go into low-level details.

(2) Notations
I had a hard time with the notations. Some notations were never actually defined and/or some notations were overloaded multiple times. Although some are very minute examples and can be sometimes understood from the context, I think the paper should be thorough in actually formally defining them.
-There are bar{\epsilon} and \hat{\epsilon} used to describe the price of fairness term in Section 5 that are never defined, although I suppose there are empirical estimates?
-\mathcal{F}_{EO} (in section 3.4) is actually never defined.
-\hat{f}^EO_{\lambda, \mu} is used in both equation (7) and in section 3.5 again to describe the closed form solution under different regularization terms (either unnormalized or partially normalized).
-Similarly, (9) and (10) used to describe V_{SX|Y} again. Overloading of the notations in this manner make it somewhat hard to follow.


(3) How good of a proxy are these regularization terms for the original fairness goals?
-My understanding is that the main highlight of this paper is the regularization terms that work to incentivize independence between two random variables. The only discussion regarding the appropriateness of these regularization terms in terms of the original fairness goal seems to be found in equation (13): i.e. it’s 0 if and only if they are independent. However, the actual problem has a penalty parameter mu which controls how big this value is, meaning it will most likely not be 0. In the case they are not 0, are these still good proxies for the original fairness goals (i.e. making sure the difference in false positive rates be small across groups).

---

### Review · Reviewer_EGvW · 2023-04-23

**Summary Of Contributions:**

In this paper, the authors propose a method for fair kernel regression through regularization based on the conditional independence operator. In particular, Equalized Odds (EO) is used as the fairness criterion, and multivariate protected attributes are considered. The authors also provide closed-form estimators for the resulting models and demonstrate empirical performance on several benchmark fairness datasets.

**Audience:**

Yes

**Claims And Evidence:**

No

**Requested Changes:**

1. Does adding the additional independence inducing regularization violate the representer theorem? In particular, why the optimal estimator still has a linear form?
2. How does the regularization coefficient impact the fairness-accuracy tradeoff?
3. For multivariate S with discrete values, the RKHS kernel for S becomes more complicated. How does the method address this setting and how does it perform in practice?

**Strengths And Weaknesses:**

[Strengths]
1. The paper includes a comprehensive summary of related work.
2. The ideas of the paper are clearly presented, fair regression is achieved by introducing regularization that induces independence between the response and the protected attributes.
3. The authors consider two independence inducing regularizations to achieve fairness, one uses the well-known HSIC as the regularization and the other adapts the conditional covariance operator. For both formulations, the authors provide closed-form estimators for the model. However, it should be noted that the independence conditions are not new, and the real contribution is the adaptation of the conditional covariance operator in fair regression.

[Weaknesses]
1. The presentation could be improved by compressing the background context (the main contribution starts at Sec 4).
2. The contribution of the paper seems incremental --- both the HSIC and the conditional covariance operator have been widely studied in literature, and the HSIC based regularization has already been proposed for fairness applications, e.g., Pérez-Suay et al., 2017 proposed the HSIC for fair kernel regression. Additionally, the independence criterion based on conditional covariance operators (Eq 6 and Eq 9) was introduced by Fukumizu et al., 2004, 2008.
3. Regarding the theory, adding another regularization based on the conditional covariance operator can violate the representer theorem, i.e., the solution may no longer have a nice linear form. I think this issue is not addressed in the paper.
4. The impact on the fairness-accuracy tradeoff has not been discussed, In particular, it is trivial to obtain a fair estimator, e.g., a predictor that outputs random noise, choosing the covariance kernels for the protected attributes and the response is subtle. Additionally, the regularization coefficient in the paper can have a significant impact on the tradeoff. I think this needs to be discussed in more detail.

---

### Review · Reviewer_AZ5T · 2023-05-02

**Summary Of Contributions:**

This paper studies the problem of imposing demographic parity (DP) and equalized odds (EO) fairness criteria in a general kernel regression problem. To do so, the authors propose to incorporate a penalization term in the loss function that captures the degree of dependence between the predictor and sensitive attributes (DP) or the predictor and sensitivity attributes conditionally on the outcome (EO). For DP, the authors use the Hilbert-Schmidt independence criterion and for EO, the conditional cross-covariance operator (which measures independence using reproducing kernels).

By modeling the problem as a nonparametric regression problem in an RKHS, the authors are able to obtain closed-form expressions for the best fair prediction function satisfying the constraints -- see equation (5) and (7). Furthermore, they can obtain closed-form bounds on the ``cost of fairness' as measured by the L2(P) distance between the best fair predictor and the best unconstrained predictor (Theorem 3.2 and Theorem 3.3). For estimation, the authors implement their procedures with a gaussian process model with linear functional form assumption on the CEF.



**Audience:**

Yes

**Broader Impact Concerns:**

I have no concerns about the broader impact of this paper.



**Claims And Evidence:**

Yes

**Requested Changes:**

Please see the weakness I mentioned above.

**Strengths And Weaknesses:**

(S1) The paper leverages the tractability of RKHS' to obtain clean results that compare the L2(P)-difference between DP and EO constrained predictors and the best unconstrained predictors.
(S2) The Gaussian process implementation leads itself to a tractable estimation procedure as a simple constrained loss minimization problem.
(S3) The procedure appears to work well in the numerical experiments.

(W1) Could the main theorems be stated directly in terms that compare the loss of the best fairness constrained predictor against the best unconstrained predictor?
(W2) The implementation in terms of a linear CEF is simple. Could this be generalized to allow for possible non-linear Gaussian processes? This may complicate how exactly the fairness constraints are imposed (it doesn't seem like simple constraints would be enough). At the same time, you could also incorporate arbitrary non-linear basis functions b(X) into the linear model for generality. This should be remarked on in the paper.

---

### Decision · Action_Editors · 2023-06-16

**Recommendation:** Accept with minor revision

**Comment:**

The paper has been found interesting by the reviewers that still had some concerns. The replies to the reviewers were precise and on point and the paper was judged acceptable by majority of the reviewers but some still had some concern. The main concerns were about some claimed contributions that were already existing and need to be toned down and another one is the existence of the representer theorem. Those concerns seem to be minor but that must be addressed before final acceptance, which is why Accept with minor revision has been selected

More details below:
+ The authors need to clarify their contributions. For instance many ideas (HSIC, Cross-Covariance) have been proposed in other papers and are clearly stated as such. But the use of conditional independence that is claimed as a contribution (contribution 2)  was already proposed by Fukumizu et al. which btw is stated in the main paper.  There are enough novel results in the paper to justify acceptance without this contribution.
+ The authors must recall the assumptions for applying the represented theorem and details how it can be used on equation (3). The response to the reviewer about this question was clear but it should appear in the paper. maybe the notation $f(X)$ is unclear and should appear as $\{f(x_i)\}_i$ ?



**Audience:**

The paper addresses an important problem (fairness) and brings novel formulations (and closed form solutions). It is definitively of interest to the audience of TMLR.

**Claims And Evidence:**

The claims in the paper are for the most part supported (by theory and experiments). The method is based on existing works but is still novel. It was noted that the paper needs some work on restating the contributions and some more details that will be discussed more in the comments below.